# Improvement of Peri-Implant Repair in Estrogen-Deficient Rats Fed a Cafeteria Diet and Treated with Risedronate Sodium

**DOI:** 10.3390/biology11040578

**Published:** 2022-04-11

**Authors:** Ana Cláudia Ervolino da Silva, Fábio Roberto de Souza Batista, Jaqueline Suemi Hassumi, Letícia Pitol Palin, Naara Gabriela Monteiro, Paula Buzo Frigério, Roberta Okamoto

**Affiliations:** 1Department of Diagnosis and Surgery, Araçatuba Dental School, São Paulo State University Júlio de Mesquita Filho—UNESP, Aracatuba 16015050, SP, Brazil; fabiorsbatista@gmail.com (F.R.d.S.B.); jaquelinehassumi@hotmail.com (J.S.H.); leticiappalin@gmail.com (L.P.P.); naaragmonteiro@gmail.com (N.G.M.); paula.frigerio@outlook.com (P.B.F.); 2Department of Basic Sciences, Araçatuba Dental School, São Paulo State University Júlio de Mesquita Filho—UNESP, Aracatuba 16015050, SP, Brazil; roberta.okamoto@unesp.br

**Keywords:** osteoporosis, ovariectomy, metabolic syndrome, risedronic acid

## Abstract

**Simple Summary:**

Postmenopausal osteoporosis, characterized by an imbalance in the remodeling cycle in which bone resorption exceeds bone formation, affects a large part of the population seeking rehabilitation with osseointegrated implants, making the prognosis of these patients unfavorable. It is estimated that approximately 99 million people over the age of 50 were diagnosed with osteoporosis in the United States in 2010. A range of drugs are used for the treatment of postmenopausal osteoporosis, aiming to prevent skeletal fractures in individuals with this osteometabolic disorder. Bisphosphonates are widely prescribed drugs to increase bone mineral density (BMD) and decrease the risk of skeletal fractures in patients with osteoporosis, with good results in this regard. However, little attention has been paid to the impact that the mechanism of action of this drug generates on the bone repair process, and more scientific evidence is needed to better understand the role of this drug in the peri-implant repair process.

**Abstract:**

(1) Background: Postmenopausal osteoporosis combined with an unhealthy lifestyle can lead to the development of metabolic syndrome, a common condition in individuals requiring oral rehabilitation. Bisphosphonates are used to increase bone mineral density. However, further studies are needed to evaluate the action of this drug on the bone repair process in the jaws. The aim of this study was to evaluate the peri-implant repair of rats with estrogen deficiency and metabolic syndrome treated with risedronate sodium. (2) Methods: Twenty-four female Wistar rats were divided into three groups: SHAM: sham surgery; OVX/SM: ovariectomy combined with a cafeteria diet; OVX/SM/RIS: ovariectomy associated with a cafeteria diet and treatment with sodium risedronate. After 30 days, the animals underwent extraction of the upper first molars. Thirty days after the extraction, an implant was installed in the same region. Sixty days after the implant was installed, the animals were euthanized for biomechanical analysis and confocal microscopic analysis. After confirming the normal distribution of the sample data, a one-way ANOVA test was performed, followed by Tukey’s post-test, with a 5% significance level. (3) Results: Significant bone preservation was observed in the risedronate-treated group. Higher removal torque values were obtained by the risedronate-treated group. (4) Conclusions: Better biomechanical performance of the implants installed in the animals treated with risedronate sodium was observed.

## 1. Introduction

Osteoporosis is a multifactorial systemic skeletal disease characterized by progressive bone loss caused by unbalanced and/or uncoupled bone remodeling activity [1]. As a consequence, deterioration of the bone microarchitecture is observed, which can lead to the occurrence of fractures at sites such as the femur, hip, and vertebrae [2]. These fractures are frequently associated with increased morbidity and mortality [3,4].

In the postmenopausal period, the decrease in estrogen levels causes a series of metabolic changes that impact many biological processes [5]. Estrogen deficiency is considered to be the primary factor in the reduction in bone mass and deterioration of skeletal structure that can lead to the development of osteoporosis [6]. According to the International Osteoporosis Foundation (2009) [7], treatment of the disease is a public health priority. The disease affects up to one-third of women over the age of 50, who have an imminent risk of suffering some type of bone fracture [7].

The hormonal changes caused by menopause lead to the appearance of lipidic metabolic disorders [5]. Postmenopausal women present a higher percentage of fat and abdominal fat accumulation [8]. This predisposition, added to the unhealthy eating habits of the current population and low physical activity, favors the appearance of metabolic syndrome, a set of interdependent factors such as insulin resistance, obesity, dyslipidemias, and hypertension, including cardiovascular diseases. Studies show a correlation between osteoporosis and metabolic syndrome [9,10,11], with both conditions being present in much of the population. However, further studies are needed to better characterize this correlation [10]. 

In their review, Tella and Gallagher (2014) [12] conducted research on the types of treatment for osteoporosis, which can be through the use of pharmacological or non-pharmacological agents. Pharmacological therapy can be established through antiresorptive and anabolic agents. Bisphosphonates, selective estrogen receptor modulators (SERMS), and denosumab constitute the group of antiresorptive agents. Parathyroid hormone (PTH) belongs to the group of anabolic agents, which promote bone formation [13]. Despite the wide range of commercially available options, drugs classified as antiresorptive in the bisphosphonate class remain the first choice for anti-osteoporosis therapy [14]. Clinical data show that its antiresorptive response is unquestionable, with good results in preventing osteoporotic fractures. However, in dentistry, there are still concerns about its use, mainly due to the occurrence of osteonecrosis of the jaws caused by antiresorptive drugs, and, among them, bisphosphonate occupies a prominent position [15]. Therefore, in this study, we investigated the action of sodium risedronate, a third-generation nitrogenated bisphosphonate with potent antiresorptive action, on the peri-implant repair process in rat maxilla.

## 2. Materials and Methods

### 2.1. Experimental Design

To mimic systemic changes, rats (Rattus novergicus albinus, Wistar) underwent ovariectomy surgery and were fed a cafeteria diet, and risedronate sodium was systemically administered as an anti-osteoporosis medication. The Lee index control was performed periodically (days 0, 60, and 90) to follow the development of metabolic alterations. Biomechanical (removal torque) and dynamic histometric (confocal microscopy) analyses were performed based on samples obtained at 60 days after implantation. At the beginning of the experiment, the selected rats underwent sham surgery or bilateral ovariectomy surgery (day 0). After recovery from surgery, the animals received the cafeteria diet to induce the development of metabolic syndrome (day 0). After 30 days (day 30), drug treatment with risedronate sodium (0.35 mg/kg/week) [16] or vehicle (physiological solution of 0.% sodium chloride) via gavage was started. At 30 days after the start of drug treatment (day 60), the upper first molars of all animals were extracted. Thirty days after extraction, osseointegrated implants were placed in the maxilla of these animals (day 90). At 14 and 42 days after implant installation, the animals received fluorochrome calcein (day 104) and alizarin (day 132) injections, respectively.

At 60 days after implant installation (day 150), the animals were euthanized to measure removal torque, and the jaws were collected for dynamic histometric analysis of the peri-implant repair (Figure 1).

### 2.2. Animals

After approval by the Ethics Committee on Animal Use (CEUA) of the Araçatuba School of Dentistry (UNESP) (protocol 00434-2019), the proposed experiments were started. In this study, 24 female Wistar rats (Rattus novergicus albinus) from the central animal house of the Araçatuba Dental School (UNESP) were used. At the beginning of the experiment, the rats were 4 months of age, and their body weight was about 250 g. The animals were kept in climate-controlled cages with a 12 h light/dark cycle and received water ad libitum. To induce the metabolic syndrome model, after adapting to environmental conditions and receiving a balanced diet and water ad libitum for a week in the faculty’s facilities, the rats began to receive a cafeteria diet, as is described in the following, with water ad libitum being maintained. Twenty-four rats were divided into three experimental groups: (1) SHAM group (*n* = 8), in which the animals underwent a sham ovariectomy and received a balanced diet and gavage with a physiological solution of 0.% sodium chloride. (2) OVX/SM group (*n* = 8), in which the animals underwent ovariectomy and received a balanced diet and gavage with a physiological solution of 0.9% sodium chloride. (3) OVX/SM/RIS group (*n* = 8), in which animals underwent ovariectomy surgery and received a cafeteria diet and drug treatment with risedronate sodium.

### 2.3. Cafeteria Diet

For animals in the OVX/SM and OVX/SM/RIS groups, a high energy density diet (cafeteria diet) was provided throughout the experiment, following the models of Gomez-Smith et al. (2016) [17] and Carillon et al. (2013) [18]. For each animal, 30 g of food was provided daily, which varied between stuffed crackers, wafer-type crackers, and corn snacks (Table 1). In addition, 50 mL of 12% sucrose solution was provided daily to each animal. The SHAM animals were fed a balanced diet (Nuvilab, Quimtia^®^, Curitiba, Brasil) (Table 2).

Weight and length, according to naso-anal measurements, were obtained periodically (days 0, 60, and 90) and allowed the calculation of the Lee index. The Lee index consists of the ratio of the cube root of the weight in grams to the naso-anal length in centimeters multiplied by 1000. Animals with a Lee index above 300 were considered obese [19].

### 2.4. Ovariectomy

After anesthesia with xylazine hydrochloride (Xilazine-Coopers, Brazil, Ltd., Osasco, SP, Brazil) and ketamine hydrochloride (ketamine hydrochloride, injectable, Fort Dodge, Animal Health, Ltd., Campinas, SP, Brazil), the animals were immobilized on a surgical board in lateral decubitus a 1 cm incision was made in the flanks, and the subcutaneous tissue and then the peritoneum were divided by layers to access the abdominal cavity. The ovaries and uterine horns were then located and lacquered with Polyglactin 910 4.0 wire (Vicryl 4.0, Johnson & Johnson, New Brunswick, NJ, USA). After that, the ovaries were removed. For synthesis, suturing was performed in layers with Polyglactin 910 4.0 thread (Vicryl 4.0, Johnson & Johnson, New Brunswick, NJ, USA). The rats in the SHAM group underwent the same procedure, but only surgical exposure of the uterine horns and ovaries was performed, without their respective ligation and removal.

### 2.5. Drug Treatment with Risedronate Sodium

Sodium risedronate was administered to the animals in the OVX/SM/RIS group at a dose of 0.7 mg/kg, according to the instructions on the package insert. Drug treatment was started 30 days after exodontia surgery and was maintained throughout the experiment. Using a curved cannula (Insight equipment), the drug diluted in physiological solution 0.9% was administered weekly. The gavage cannula was used to bring the medication closer to the esophageal region. The final volume administered was 0.3 mL. The same amount of the vehicle (physiological solution 0.9%) was administered to the rest of the groups.

### 2.6. Tooth Extraction

The animals were fasted for eight hours before the surgical procedure, sedated by the combination of 50 mg/kg intramuscular ketamine (Vetaset-Fort Dodge Saúde Animal Ltd., Campinas, SP, Brazil) and 5 mg/kg xylazine hydrochloride (Dopaser-Laboratório Calier do Brasil Ltd., Osasco, SP, Brazil), and received mepivacaine hydrochloride (0.3 mL/Kg, Scandicaine 2% with adrenaline 1:100,000, Septodont, SaintMaur-des-Fossés, France) as local anesthesia and for hemostasis of the operative field.

After sedation, the animal was positioned on a surgical table specially made for surgical procedures in rodents, favoring the maintenance of an open oral cavity and providing an adequate positioning for the exodontia procedure. After antisepsis of the area using topical polyvinylpyrrolidone iodine (PVPI 10%, Riodeine Degermante, Rioquímica, São José do Rio Preto, SP, Brazil), the extraction of the upper first molar was performed with the aid of a Hollenbeck 3S sculptor (Quinelato, Rio Claro, SP, Brazil). Diuresis, syndesmotomy, and then dislocation of the upper first molar were performed with the Hollenbeck 3S sculptor using the wedge and lever principle, and then elevation of the dental element (exodontia) was performed. Finally, the alveolar mucosa was sutured with Polyglactin 910 4.0 thread (Figure 2A–D) (Vicryl 4.0, Johnson & Johnson, New Brunswick, NJ, USA).

### 2.7. Maxillary Implant Installation

Thirty days after extraction of the first upper molar, the osseointegrated implants were installed in the maxilla of the animals. With a number 15 blade (Feather Industries Ltd., Tokyo, Japan), a linear incision of approximately 0.5 cm length was made on the post-exodontic alveolar ridge (Figure 3A), and then the soft tissue was detached in full thickness and pulled apart with the aid of a Hollenbeck sculptor, exposing the bone tissue for implant installation.

Twenty-four commercially pure titanium grade IV implants were installed based on the concept of double acid etching (Emfils Comércio Produtos Odontológicos, Itu, SP, Brazil), with a diameter of 1.4 mm and height of 2.7 mm, sterilized by gamma rays. For this, drilling was performed with a 1.2 mm diameter spiral drill mounted on an electric motor (BLM 600^®^; Driller, São Paulo, SP, Brazil) at a speed of 1000 rpm under irrigation with isotonic 0.9% sodium chloride solution (Fisiológico^®^, Laboratórios Biosintética Ltd., Ribeirão Preto, SP, Brazil) and counter-angle with 20:1 reduction (Angle piece 3624N 1:4, Head 67RIC 1:4, KaVo^®^, Kaltenbach & Voigt GmbH & Co., Biberach, Baden-Württemberg, Germany) and depth 2.0 mm, with locking and initial stability (Figure 3B).

Each animal received 1 implant in the same region of the right upper first molar (Figure 3C–E). Monofilament yarn (Nylon 5.0, Ethicon, Johnson, São José dos Campos, SP Brazil) with interrupted stitches was used for tissue closure (synthesis) (Figure 3F).

In the immediate postoperative period, each animal received a single intramuscular dose of 0.2 mL of penicillin G-benzathine (Small Veterinary Pentabiotic, Fort Dodge Saúde Animal Ltd., Campinas, SP, Brazil). The animals were kept in cages throughout the experiment with feed and water ad libitum.

### 2.8. Analysis

#### 2.8.1. Biomechanical Analysis

Sixty days after implant installation, the animals were sedated with a combination of 50 mg/kg intramuscular ketamine (Vetaset-Fort Dodge Saúde Animal Ltd., Campinas, SP, Brazil) and 5 mg/kg xylazine hydrochloride (Dopaser-Laboratório Calier do Brasil Ltd., Osasco, SP, Brazil). The maxillae were accessed for implant exposure and reverse torquing. An implant holder (Neodent, Curitiba, PR, Brazil) was adapted to the hexagon of the implant, and a digital torque meter was attached to the holder. A counterclockwise movement was applied, increasing the reverse torque until the implant rotated within the bone tissue, completely rupturing the bone/implant interface, at which time the torque wrench recorded the maximum peak torque for the rupture in Newton per centimeter (N.cm). 

#### 2.8.2. Dynamic Histometry

For dynamic histometry, at 14 and 42 days after implant installation, the animals received 20 mg/kg calcein and 30 mg/kg alizarin, respectively. Both substances are fluorochromes that have affinity for calcium during the precipitation process on the bone matrix. The collected samples (eight maxillae per group) were dehydrated in an increasing sequence of alcohols: 70%, 90%, and 100%. The specimens were then immersed in a methyl methacrylate (MMAL) solution (Classical, Classic Dental Articles, São Paulo, SP, Brazil), followed by three MMAL baths. Benzolium peroxide catalyst (1%, Riedel de Haën AG, Seelze-Hannover, Lower Saxony, Germany) was added to this last bath. The parts were placed in a test tube filled with the solution and kept in an oven at 37 °C for 5 days until final polymerization of the resin. After polymerization, the test tubes were broken, and the resin was longitudinally worn with the aid of a MaxiCut mounted on a bench motor (Kota-São Paulo-SP, Brazil). The pieces were then bilaterally worn with increasing grain sizes of 120, 300, 400, 600, 800, and 1200 mounted on an automatic polishing machine (ECOMET 250PRO/AUTOMET 250. Buehler, Lake Bluff, IL, USA) until the cuts reached a thickness of 80 μm. A digital caliper was used for measurement (Mitutoyo, Pompéia, SP, Brazil). The slices were then mounted on glass slides with mineral oil (liquid petroleum, Mantecor, Taquara, RJ, Brazil) and sealed with a glass cover and enamel to avoid oil leakage and possible dehydration of the specimen. They were evaluated longitudinally in the region of the bone/implant interface corresponding to the third, fourth, and fifth rows of the implants. These slices were captured with a Leica CTR 4000 CS SPE (Leica Microsystems, Heidelberg, Baden-Württemberg, Germany) confocal laser microscope using a 10× objective (original magnification 100) at the Bauru Dental School-USP. Thus, calcein and alizarin red fluorochrome images (old bone/new bone) were used to evaluate bone and the overlap of the two images in terms of mineral apposition rate (MAR) and bone dynamics. The images were analyzed in ImageJ software (Image Processing and Analysis Software, Bethesda, MD, USA). Using the Hands-Free Tool, the areas of fluorochrome precipitation (calcein/alizarin) were measured. With the Straight tool, the MAR was found through five measurements extending from the outer margin of calcein toward the outer margin of alizarin. The value obtained was divided by 28, which represents the interval of days between the injections of the two fluorochromes analyzed. For bone dynamics, using the same tool, the total area of each fluorochrome was measured.

#### 2.8.3. Statistical Analysis

The values obtained from the analyses were submitted to statistical analysis using GraphPad Prism 7.01 software (GraphPad Software, San Diego, CA, USA). To select appropriate statistical data, tests for homogeneity and homoscedasticity (Shapiro–Wilk) were performed. After confirming normal distribution, one-way ANOVA test was performed to compare the results obtained in SHAM, OVX/SM, and OVX/SM/RIS groups, followed by Tukey’s post-test when necessary. A 95% significance level was adopted (*p* < 0.05).

#### 2.8.4. Sample Size

The sample size was determined from the sample calculation by a power test performed at http://www.openepi.com/SampleSize/SSMean.htm (OpenEpi, version 3, open-source calculator; accessed on 7 March 2022). We obtained the mean number (*n*) of animals for this study using the study published by Abtahi et al. 2013 [20], in which data regarding counter-torque were used, with the mean used for group X = 11.1 and for group Y = 7.4 and the standard deviation for X = 3.1 and Y = 1.9, with a significance level of 5% and power of 95% in a one-tailed hypothesis test; the total sample size was the same as that used in the present study. The total number of samples used in this study was 16 per group, *n* = 8 animals per group, with bilateral implant installation surgery, totaling 16 samples. The samples were randomly distributed between biomechanical analysis (counter-torque) and laser confocal microscopy analysis.

## 3. Results

### 3.1. Lee Index

The cafeteria diet contributed to the development of obesity, reflected in higher Lee index values in the OVX/SM and OVX/SM/RIS groups at the end of the experiment (Table 3).

### 3.2. Biomechanical Analysis

Data obtained by reverse torque analysis showed that the group with systemic alterations (OVX/SM) had a lower removal torque value when compared with the SHAM group, but the difference was not statistically significant (SHAM = 3.8 N.cm versus OVX/SM = 3.3 N.cm, *p* = 0.79). When we observed the removal torque values of the OVX/SM/RIS group, we noted significant improvements in biomechanical parameters, which are reflected in the higher removal torque values of this group, showing statistically significant differences compared to the other groups (SHAM = 3.8 N.cm versus OVX/SM/RIS = 6.8 N.cm, *p* = 0.008; OVX/SM = 3.3 N.cm versus OVX/SM/RIS = 6.8 N.cm, *p* = 0.001) (Figure 4). The reverse torque data show that risedronate sodium treatment significantly improves peri-implant bone tissue quality when compared to animals with estrogen deficiency/metabolic syndrome and even when compared to animals in normal health conditions (SHAM animals).

### 3.3. Dynamic Histometry

Similar values of the mineral apposition rate were observed in the three experimental groups, indicating that there was no difference in mineral precipitation during peri-implant repair between animals in healthy conditions (SHAM group) and those with estrogen deficiency and metabolic syndrome (OVX/SM). Treatment with risedronate sodium did not interfere with mineral precipitation in the OVX/SM/RIS group (Figure 5). 

As for bone dynamics, in the SHAM group, there was a balance in the precipitation of the fluorochromes calcein and alizarin, evidencing the physiological conditions and homeostasis of the peri-implant repair process in animals in this group (SHAM: calcein = 2070 μm; alizarin = 1294 μm²). In the OVX/SM group, a slightly higher amount of the fluorochrome alizarin was found when compared to calcein, suggesting a delay in the maturation and mineralization process of this tissue, but without statistical differences (OVX/SM: calcein = 1147 μm²; alizarin = 2299 μm²). In the OVX/SM/RIS group, significantly higher values of calcein area were obtained, which were statistically different from those in all other groups and even from alizarin in the same group, indicating a decrease in bone turnover in this group and greater preservation of “old bone”, attributed to lower bone resorption promoted by risedronate sodium (OVX/SM/RIS: calcein = 7485 μm²; alizarin = 4216 μm²) (Figure 6).

## 4. Discussion

In the present study, it was observed that risedronate sodium improved osseointegration responses based on the evaluation of removal torque after implant installation in rats with estrogen deficiency associated with metabolic syndrome. In relation to bone remodeling activity, risedronate was found to have effective antiresorptive characteristics, without impairing the quality of the bone that maintained resistance to the removal torque test.

Despite the high survival rates of osseointegrated implants, failures and complications of this type of rehabilitation still occur [21]. To achieve successful rehabilitative treatment with dental implants, the occurrence of osseointegration is of fundamental importance. Osseointegration is strongly influenced by local and systemic factors, such as the bone quantity and quality in the bone bed and homeostasis of bone metabolism. Some osteometabolic disorders can compromise bone homeostasis and thereby also compromise osseointegration [22]. 

Osteoporosis is an important public health issue, affecting about 200 millions people worldwide. This condition is characterized by an imbalance in the remodeling process with excessive rates of bone resorption, resulting in a continuous decrease in bone volume and quantity and deterioration of bone tissue microarchitecture, leading to decreased primary stability of implants and consequently lower bone/implant contact (BIC). To evaluate the peri-implant repair process under the influence of estrogen deficiency, an experimental model of bone loss after menopause was used. This model is characterized by bone loss after bilateral ovariectomy and is the most widely used model to promote osteopenia in adult rats and, for this reason, was selected as the method of the present study [23,24].

Treatment of osteoporosis with risedronate sodium, which is known to act by inhibiting bone remodeling, is widely administered with good results with regard to increasing bone mineral density and preventing fracture risk [25,26]. However, the use of this drug can have adverse effects, such as impaired osseointegration responses after implant installation in the jaws, and can lead to the development of osteonecrosis of the jaws. Although it is discussed rather superficially by the medical profession, it has haunted implant dentistry professionals [9,15]. On the other hand, the importance of maintaining the use of these antiresorptive drugs, which are the first choice for the treatment of osteoporosis, has caused conflicts between the medical and dental classes for some time now. Currently, there are studies in the literature that discuss the “drug holiday” and the constant monitoring of patients associated with the predictive tool called FRAX (a tool used to assess the fracture risk of patients) in order to try to more accurately monitor the effects of bisphosphonates (the class of medications most frequently used today) on bone tissue, considering their half-life and their capability of remaining incorporated into the mineralized bone matrix. It is worth noting that interruptions in the use of these antiresorptive drugs can lead to risks of vertebral and hip fractures, which can lead to a series of complications, including death [25]. 

The increase in life expectancy, the search for rehabilitative procedures using osseointegrated implants, and the need to provide better conditions and greater access of the population to rehabilitative treatments with implants make this topic very current and relevant. Thus, our goal was to evaluate the effect of risedronate sodium, a bisphosphonate widely prescribed to treat postmenopausal osteoporosis, on the peri-implant repair process in an experimental model that mimics a very common condition in postmenopausal women who, as a consequence of estrogen deficiency, start to develop changes in lipid metabolism that can evolve to metabolic syndrome. For this study, the in vivo model chosen was the maxillary implant, which we considered essential for mimicking the situation found in dental practice as much as possible. Therefore, the animals have an estrogen deficiency, which is known to be a condition that predisposes to osteoporosis. However, one of the alterations observed with estrogen deficiency and the consequent menopause in women consists of alterations in the lipid profile and dyslipidemias, which consequently may contribute to the occurrence of metabolic syndrome. In the present study, we aimed to get as close as possible to this condition using a cafeteria diet, considering that estrogen deficiency and metabolic syndrome are two situations commonly observed in osteoporotic patients and are interrelated [27].

In other studies developed in our laboratory in which we compared the absence or presence of metabolic syndrome in experimental animals, we observed that it also plays an important role in bone tissue metabolism. There are studies in the literature that discuss controversial results on the effect of metabolic syndrome on bone density [28,29]. Recent findings have shown that white adipose tissue is responsible for the secretion of adipokines, such as leptin and adiponectin, which are closely related to metabolic diseases such as obesity, insulin resistance, and diabetes and play an important regulatory role in bone diseases [30]. Evidence suggests that leptin and adiponectin have a direct anabolic effect on osteoblasts and act indirectly to elevate levels of alkaline phosphatase, osteocalcin, and type 1 collagen [31]. However, when associated with aging and hypoestrogenism, these same adipokines may have a negative effect on bone metabolism by increasing proinflammatory cytokines such as TNF and IL-6 and favoring osteoblast apoptosis and osteoclast proliferation [32]. We intend to investigate this interference with the lipid profile and bone tissue metabolism in more detail.

It is worth pointing out that one of the limitations of this study was the characterization of the metabolic syndrome model, since in order to prove its development, it would be necessary to have a trained professional to periodically collect blood from the experimental animals in order to determine inflammatory markers, cholesterol increase, and obesity-related enzymes. Another important aspect is the sensitivity of the technique for installing implants in the maxilla, since the installation site is next to noble structures, and extreme caution is required during surgery to provide safety and comfort to the animals.

It is worth noting that risedronate was the bisphosphonate chosen for this study, since it is widely used by the population, along with alendronate sodium. As our group has already studied the effects of the latter on alveolar and peri-implant repair [33,34], we considered that it would be interesting for another bisphosphonate to be chosen for this experimental design.

Metabolic changes had a slight influence on the biomechanical stability of the implants installed in the animals in the OVX/SM group, with removal torque values slightly lower than those obtained by the SHAM group but without a statistical difference. In the same group, higher bone turnover was evidenced by lower calcein precipitation (old bone) and slightly higher alizarin precipitation (new bone), indicating greater resorption activity that was probably induced by the metabolic changes developed in the animals. Risedronate treatment had positive effects on the biomechanical parameters of implants installed in animals with compromised systemic conditions, which are reflected in a significant increase in implant removal torque values in animals in the OVX/SM/RIS group. This finding alone is very interesting as to the choice of this bisphosphonate. It is worth noting that in studies performed by our group, when removal torque was evaluated using sodium alendronate [3], the values obtained were very low and very close to those observed in animals with osteoporosis. This aspect was not confirmed in the present study, as risedronate treatment showed better results when compared to animals with systemic involvement. Proving the antiresorptive effect of risedronate sodium, we observed dramatically higher calcein values in the OVX/SM/RIS group when compared to the other experimental groups. Presumably, the presence of “old bone” provided an adequate bone bed and promoted better primary stability conditions for osseointegrated implants, which generated higher removal torque values. 

According to these results, we can conclude that risedronate sodium showed a favorable clinical performance in the peri-implant repair process. This leads us to infer that there is a difference in the dynamics of its mechanism of action when compared to other bisphosphonates that impair the bone repair process [3]. However, further studies are needed to better characterize the action of this drug in the process of peri-implant repair. 

## 5. Conclusions

From the results obtained in this study, it can be concluded that risedronate sodium had an important antiresorptive effect during the peri-implant repair process, promoting a decrease in bone turnover and consequently the maintenance of pre-existing bone tissue, which contributed to an improvement in biomechanical parameters based on the evaluation of the removal torque of implants installed in animals under treatment with this drug. Clinical studies are needed to better understand the relationship between osteoporosis and dyslipidemia and the possible impact of these changes on peri-implant repair using antiresorptive drugs.

## Figures and Tables

**Figure 1 biology-11-00578-f001:**
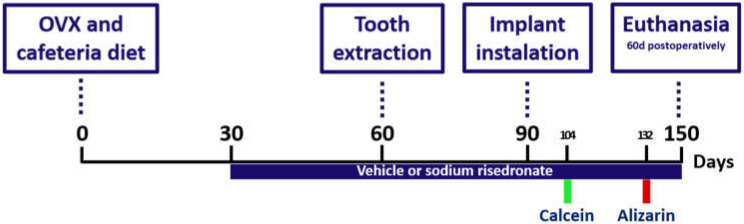
Scheme illustrating the experimental design according to the procedures performed in the study.

**Figure 2 biology-11-00578-f002:**
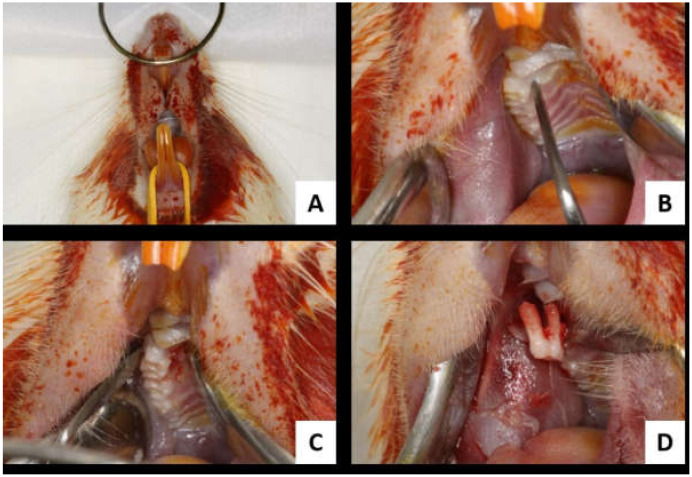
Tooth extraction. (**A**): operative site antisepsis; (**B**): syndesmotomy; (**C**): upper first molar dislocation; (**D**): first upper first molar extraction.

**Figure 3 biology-11-00578-f003:**
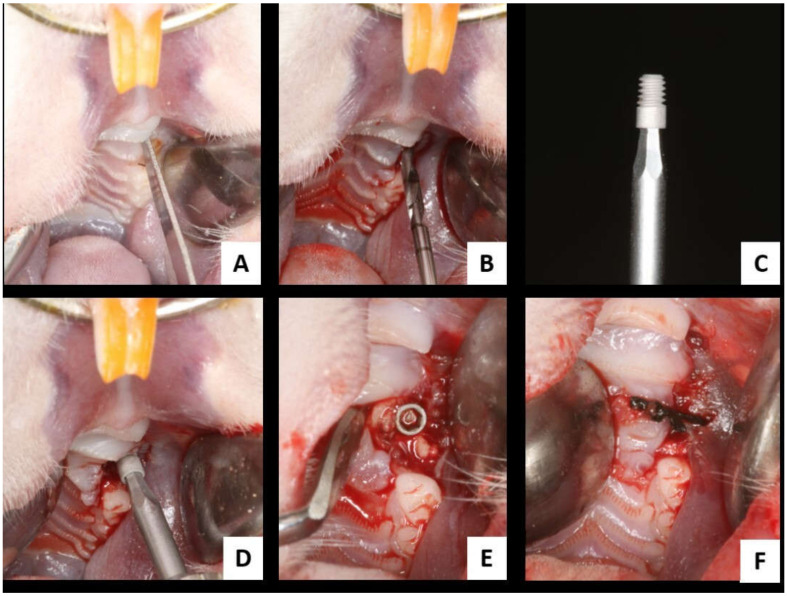
Surgery for installing osseointegrated implants in the maxilla. (**A**) Linear incision on the alveolar ridge of the upper right first molar; (**B**) alveolar bone milling cutter with diameter 1.2 mm; (**C**) implant used in the study; (**D**) osseointegrated implant installation in the maxilla; (**E**) implant installed in the maxilla; (**F**) alveolar mucosa suture.

**Figure 4 biology-11-00578-f004:**
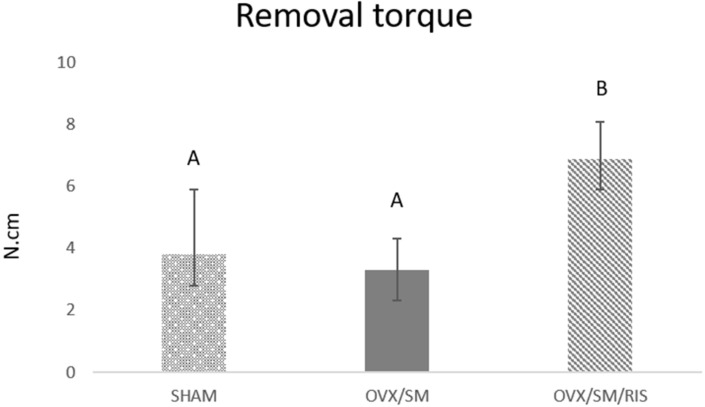
Removal torque values of SHAM, OVX/SM, and OVX/SM/RIS groups in N.cm. Different letters in the columns indicate statistical difference (*p* < 0.05, Tukey).

**Figure 5 biology-11-00578-f005:**
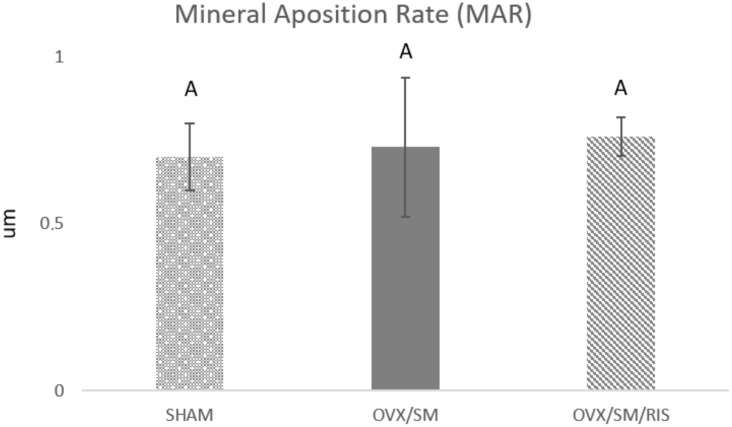
Linear evaluation of the mineral apposition rate per day (μm) of SHAM, OVX/SM, and OVX/SM/RIS groups (*p* < 0.5, Tukey).

**Figure 6 biology-11-00578-f006:**
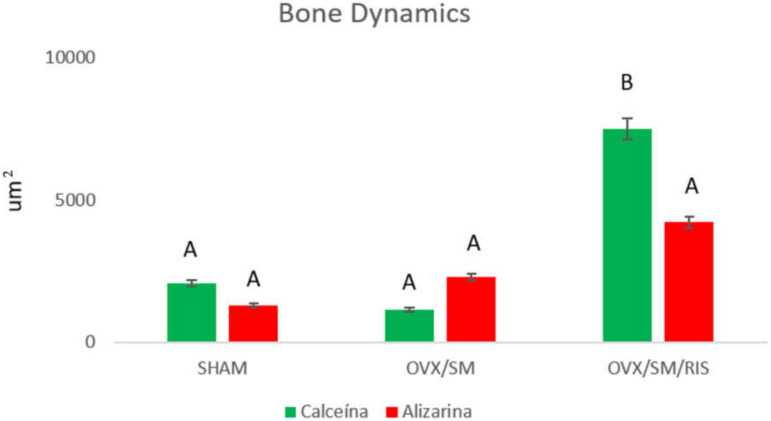
Peri-implant bone area (μm²) of the groups expressed by calcein and alizarin (*p* < 0.05, Tukey). Distinct letters (A or B) indicate statistical difference (*p* < 0.05).

**Table 1 biology-11-00578-t001:** Cafeteria diet composition for OVX/SM and OVX/SM/RIS groups.

Food Item	Kcal	Tot. Fat (g)	Carb. (g)	Prot. (g)	Ca (mg)	Sod (mg)	Zn (mg)	Pot. (mg)	Iron (mg)
Stuffed Cracker (10 g)	46	0.7	6.3	0.7	50	27.3	0.1	0.0	0.0
“Wafer” Cracker (10 g)	46	1.7	6.8	0.6	2.3	13.7	0.1	24.1	0.2
Corn Snacks (10 g)	30	1.8	6.8	0.6	0.0	39.6	0.0	33.6	0.0
Sugar Water 12% (50 mL)	30	0.0	6.0	0.0	0.0	0.0	0.0	0.0	0.0
Total	152	4.2	25.9	1.9	52.3	80.6	0.2	57.7	0.2

**Table 2 biology-11-00578-t002:** Balanced diet composition for SHAM animals.

Nutrition Information	Amount per Serving (30 g)
Kcal	113
Total fat (g)	1.3
Carb. (g)	18
Protein (g)	7.3
Ca (mg)	360
Sodium (mg)	81
Zinc (mg)	3.3
Potassium (mg)	270
Iron (mg)	5.4

**Table 3 biology-11-00578-t003:** Lee index values (mean ± standard deviation) of SHAM, OVX/SM, and OVX/SM/RIS on days 0, 60, and 90.

Evaluation Period	SHAM	OVX/SM	OVX/SM/RIS	*p* Value
Day 0	280 ± 4.9	281 ± 5.75	275 ± 4.5	*p* = 0.7
Day 60	279 ± 4.2	296 ± 7.9	300 ± 10.2	SHAM vs. OVX/SM < 0.0001 * SHAM vs. OVX/SM/RIS < 0.0001 *
Day 90	279 ± 5.0	302 ± 11.5	305 ± 12	SHAM vs. OVX/SM < 0.0001 * SHAM vs. OVX/SM/RIS < 0.0001 *

* *p* < 0.05.

## Data Availability

Data sharing not applicable.

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
