# Peer review of "Improvement of Peri-Implant Repair in Estrogen-Deficient Rats Fed a Cafeteria Diet and Treated with Risedronate Sodium"

_biology, 2022, doi:10.3390/biology11040578_

Round 1

Reviewer 1 Report

The work submitted for evaluation was entitled "Osseointegration improvement in rats with estrogen deficiency and metabolic syndrome treated with risedronate sodium". These are extremely important studies with a cognitive as well as application aspect. I believe that this work may be published after responding to detailed comments.

Detailed comments:

  1. The research hypothesis should be re-formulated and detailed. The reader may get the impression that it was introduced into the manuscript by mistake in its current form and that it concerns a different work.
  2. How was the minimum number of animals in the experimental groups determined?
  3. Despite the reference to earlier works, please state the energy value of the diet used in your research. Despite the methodological similarities, it may differ significantly. What was the share of individual components of the cafeteria diet? The same data should be provided for a balanced diet.
  4. What was the content of micro and macro elements in the diet, especially Ca?
  5. What was the concentration of the saline solution? Do the authors mean physiological solution? If so, this name should be used as more communicative.
  6. I consider the photos of the ovariectomy to be infantile and unnecessary, while the photos of tooth extraction and implant placement are amazing.
  7. What digital calliper was used to measure 80 um? Mitutoyo products have a resolution of 0.01 mm and a measuring accuracy of 0.02 mm. Such parameters do not allow for accurate measurement of 0.08 mm.
  8. What were the effects of using the cafeteria diet? There are no results of the dynamics of weight gain, lipid profile and others.
  9. Studies on osseointegration should be supported by the analysis of biochemical markers of bone tissue metabolism.
  10. Discussion line 296. "Osteoporosis is an important public health issue, affecting mainly women after menopause." I can only agree with the first half of this sentence. However, osteoporosis is also an important medical problem in men, and its complications have been associated with significant male mortality rates, greater than female mortality. I think it is worth mentioning it and spreading knowledge about it, fighting stereotypes.
  11. Discussion Line 338. It seems that the authors use the term metabolic syndrome to equate it with obesity. This is confirmed by further considerations on the increased depot of adipose tissue on bone metabolism. In the previous paragraph, a change in the lipid profile in terms of hormonal hypofunction of the ovaries was mentioned (line 333-335). Here, the reader gets the impression that the change in lipid profile after castration/during menopause is what metabolic syndrome is. This is a far-reaching simplification. Please explain on what basis the authors claim the presence of metabolic syndrome in examined rats. I have not noticed any biochemical analyzes, cardiovascular tests, etc. that would constitute the basis for the claim of the presence of the metabolic syndrome. It should be emphasized that obesity is not the same as metabolic syndrome, although it is the main cause of its development. In this aspect, please also pay attention to the title of the work.
  12. Discussion line 377. "This leads us to think that there is a difference in the dynamics of its mechanism of action when compared to other bisphosphonates with results that impaired the bone repair process." Not at all. The results of these studies do not in any way entitle the authors to infer about the dynamics. The dynamics of repair processes has not been studied.
  13. The conclusions of these studies do not correspond to the research hypothesis and require an in-depth correction.
  14. The abstract should be deeply rewritten, especially conclusions that are non-informative

Reviewer 2 Report

The study was designed to evaluate the effect of risedronate sodium on the peri-implant repair process in a rat model. The manuscript was well written but less creative for the materials and methods. I suggest a novel and thorough research such as a study in a combination with different drugs. In general, the manuscript should be improved and organized.

Reviewer 3 Report

The authors aimed to investigated the action of sodium risedronate, a third-generation nitrogenated bisphosphonate with a potent antiresorptive action, on the peri-implant repair process.

The study covers some issues that have been overlooked in other similar topics. The structure of the manuscript appears adequate and well divided in the sections. Moreover, the study is easy to follow, but some issues should be improved. Some of the comments that would improve the overall quality of the study are:

1-) The manuscript needs grammar correction. Please also check typos thorough the text;

2-) Discussion section: The authors stated that :

"To achieve successful rehabilitative treatment with dental implants, the occurrence of osseointegration is of fundamental importance. Osseointegration is strongly influenced by local and systemic factors, such as bone quantity and quality of the bone bed and homeostasis of bone metabolism. Some osteometabolic disorders can compromise bone homeostasis and thereby also compromise osseointegration.".

Reference for this paragraph are necessary. For your convenience, please see : doi: 10.7150/ijms.20522. doi: 10.2174/1871530318666180423102905; https://doi.org/10.3390/jpm12020235;

3-) Limitations of the study are missing. Please add it;

5-) Conclusion Section: This paragraph required a general revision to eliminate redundant sentences and to add some "take-home message".

Round 2

Reviewer 1 Report

I’m fully satisfied with the introduced corrections and recommend the manuscript „Improvement of peri-implant repair in estrogen-deficient rats fed a cafeteria diet and treated with risedronate sodium” for publication in BIOLOGY.

Author Response

The authors are very grateful to Reviewer 1 for his important contributions to the improvement of this study. 

Reviewer 2 Report

General comment: The study was aimed to evaluate the action of risedronate sodium on the peri-implant repair process in a rat model. The manuscript was well written. However, some revision is required before the manuscript is accepted.

Line 25 correct the sentence as “However, further studies are”

line 48 correct the sentence as “impact many biological processes”

Line 77 why a reference is cited here when you are talking about what you did in your study? It is confusing

Author Response

The authors appreciate the extremely pertinent considerations of Reviewer 2.

The sentences located on lines 25 and 48 have been corrected.

On line 77 the reference in question was inserted by mistake, and it has been removed from the manuscript.